# Equatorial wave diagnosis for the Atlantic Niño in 2019 with an ocean reanalysis.

Qingyang Song[1,2] and Hidenori Aiki[3]

[1]Key Laboratory of Marine Hazards Forecasting, Ministry of Natural Resources, Hohai University, Nanjing, China
[2]College of Oceanography, Hohai University, Nanjing, China
[3]Institute for Space-Earth Environmental Research, Nagoya University, Nagoya, Japan

**Correspondence:** Qingyang Song (qysong@hhu.edu.cn)

**Abstract.** The propagation of equatorial waves is essential for the onset of Atlantic Niño, but diagnosing waves with ocean reanalysis or in-situ data remains a challenge. This study uses an ocean reanalysis to diagnose the wave energy transfer route during the 2019 event. The climatological values and the anomaly in 2019 at each grid point are decomposed into the first four baroclinic modes based on their local density profiles. The decomposed geopotential can well reproduce the displacement of the thermocline during the event. Wave energy flux is calculated by means of a group-velocity-based scheme. In addition to detecting wind-forced Kelvin waves and reflected Rossby waves, the wave energy flux reveals another possible energy transfer route along the western boundary, where some off-equatorial wave energy can excite coastally-trapped Kelvin waves and transfer back to the equatorial Atlantic. Five transections are selected, across which the passing wave energy fluxes in 2019 are integrated. The results suggest that the Kelvin waves in the third and fourth mode are locally forced, while the wave energy in the second mode is more likely from the off-equatorial region. Therefore, in the fall of 2019, the second-mode Kelvin waves can deepen the thermocline ahead of other modes from September, serving to precondition the Niño event.

## 1 Introduction

The equatorial Atlantic Ocean is known for exhibiting pronounced anomalies of sea surface temperature (SST) on interannual time scales, of which the events with positive anomalies are often referred to as Atlantic Niños (Giannini et al., 2003). This phenomenon is essentially driven by the Bjerknes feedback, that the westerly wind anomaly caused by the SST anomaly in the eastern equatorial basin excites equatorial waves and consequently amplifies the SST anomaly through thermocline displacement due to the wave propagation. The impact of these SST events is not limited to affecting precipitation and monsoon patterns in the surrounding sea area and continent, but also extends to the Pacific and Indian Oceans through atmospheric teleconnections (Carton et al., 1996; Okumura and Xie, 2004; Rodríguez-Fonseca et al., 2009; Lübbecke and McPhaden, 2012; Foltz et al., 2019).

Since 2000, the interannual variability of SST has been revealed to become weaker (Tokinaga and Xie, 2011; Prigent et al., 2020). Although the cause to suppress the Atlantic Niño events remains unclear, the thermal dynamic rather than the oceanic dynamic was previously thought to play a critical role in the decline trend (Nnamchi et al., 2015; Crespo et al., 2022). A warmer climate deepens the upper ocean layer making it less sensitive to upwelling anomalies (Crespo et al., 2022). That is, different

from the equatorial Pacific, where the changes in the zonal SST gradient under greenhouse forcing are most relevant (Latif and Keenlyside, 2009), a basin-wide warming associated with the climate change can eventually break the classic Bjerknes loop so as to limit the development of SST anomaly. Despite the predicted long-term weakening trend for SST variability, two extremely warm SST events occurred in the winter of 2019 and the second half of 2021. As a result, a skillful prediction for those SST events is still necessary in the short term.

Evidences were reported that the Atlantic Niño events in both 2019 and 2021 are strongly related with off-equatorial Rossby waves (RW) (Richter et al., 2022). The equatorial Atlantic basin is narrow and has an inwardly-tilted coastline in the northeast. This allows waves from off-equatorial regions to travel through the basin quickly and leads more wave energy back to the equatorial sector (Song and Aiki, 2020). When the off-equatorial RWs approach the western boundary, reflected Kelvin waves (KW) are excited and consequently displace the thermocline to affect SST in the equatorial Atlantic (Foltz and McPhaden, 35 2010). This dynamic link suggests that the wave condition is still crucial for the prediction of the Niño events at this moment. Indeed, researchers have already proposed to use equatorial waves to warn the anomalous SST event in the down-wave coastal region (Imbol Koungue et al., 2017, 2019; Illig and Bachèlery, 2019). The period that wave energy transfers horizontally and vertically (normally takes several months to pass the basin depending on the baroclinic mode and around one month to affect local SST depending on the thermocline depth) from the west of the equatorial Atlantic or further from the off-equatorial region 40 gives the equatorial wave the potential to be an effective predictor for the Atlantic Niño at its onset stage.

Early warning systems based on wave propagation for the anomalous SST event in the equatorial Atlantic and the down-wave coastal regions are hence expected (Imbol Koungue et al., 2017, 2019; Song et al., 2023a). They suggested to implement ocean linear model in real time that simulates equatorial waves in addition to use in-situ and altimetric data for prediction, since RWs in off-equatorial region are previously difficult to diagnose with observed or modelled data (Rossby, 1945; Philander, 1978; 45 Schopf et al., 1981; Schiller et al., 2010). However, benefiting from a newly unified wave energy flux scheme for equatorial waves by Aiki et al. (2017), which they named as AGC flux (standing for the authors, Aiki, Greatbatch and Claus), the wave energy flux that is transferred in the tropical Atlantic Ocean can be diagnosed by only applying dataset of current velocity and pressure (Song and Aiki, 2020, 2021). Currently the high-resolution dataset (eg. 1/12° horizontal resolution and 50 vertical levels provided by Copernicus Marine Environment Monitoring Service) is available to capture the rapid gravity waves such as 50 KWs in low baroclinic mode (Jean-Michel et al., 2021); Also, Prediction and Research Moored Array in the Tropical Atlantic (PIRATA) initiated in 1997 can provide real time in-situ observation in the tropical Atlantic (Bourlès et al., 2019). These datasets offer an opportunity to develop a wave warning system that does not require a linear ocean model. This study is an attempt of applying the AGC flux in an ocean reanalysis to diagnose waveguide in the event year 2019. We hope that our investigation for the process of the wave energy transfer and its impact on the Atlantic Niño can inspire a simple and reliable 55 system to predict anomalous SST events through diagnosing existing dataset.

The structure of this manuscript is as follows: Section 2 briefly explains the applied dataset and the method to extract the wave information from the data; In Section 3 we present the decomposed waves and construct the wave energy transfer route (the trajectory of wave energy transport) in 2019, which we compare with the climatological data to evaluate the extraordinary

wave energy from both the equatorial and the off-equatorial region during the Niño event; Section 4 summarizes the addressed
problems and conclusions of this study.

## 2 Wave detection with reanalysis data

The reanalysis dataset we employed in this study is the GLORYS12V1 product from CMEMS (Copernicus Marine Environment Monitoring Service). The product is a global ocean eddy-resolving (1/12° horizontal resolution, 50 vertical levels) reanalysis using reduced-order Kalman filter and 3D-VAR scheme to involve along track altimeter data, satellite SST, sea ice concentration and in-situ temperature and salinity vertical profiles in joint assimilation (Jean-Michel et al., 2021). The product provides daily and monthly dataset including potential temperature, salinity, current velocity, mixed-layer thickness and sea surface height (SSH) covering the altimetry (1993 onward). Daily data are used to detect the equatorial waves in the event year; monthly data are applied to obtain climatology and mean ocean state. Readers can check the quality of GLORYS12V1 dataset in (Marie et al., 2022). Specifically in the tropical Atlantic, the root mean square error of temperature/salinity profiles is smaller than 0.4°C/0.3 psu; the correlation of the zonal velocity profiles between the reanalysis data and the in-situ observation from PIRATA buoys at (23°W, 0°E) is around 0.6 with the root mean square error of around 0.1 m/s in the mixed layer (10-80m) (Marie et al., 2022). Although compared with the observation, the dataset still underestimates the zonal velocity in the mixed layer by around 5%-10%, it should be proper to be applied in this study for the introduction of a diagnosis scheme.

The decomposition of baroclinic waves from the ocean reanalysis requires the solution of the eigen equation in the vertical direction as follows,

$$\frac{\partial}{\partial z}\left(\frac{1}{N^2}\frac{\partial \psi^{(n)}}{\partial z}\right) = -\frac{1}{(c^{(n)})^2}\psi^{(n)}, \tag{1}$$

which subject to the boundary conditions $\psi_z^{(n)}(0) = \psi^{(n)}(-H_b) = 0$. The symbol $H_b$ is the ocean bottom depth, $N$ is the Brunt-Väisälä frequency estimated by the mean vertical profiles of temperature and salinity, and $\psi^{(n)}$ is the eigen function of the $n$th baroclinic modes for the corresponding gravity wave speed (eigen value) $c^{(n)}$. Then based on the linear theory, the wave-induced anomaly subjects to

$$u'(x,y,z,t) = \sum u^{(n)}(x,y,t)\psi^{(n)}(z),$$
$$v'(x,y,z,t) = \sum v^{(n)}(x,y,t)\psi^{(n)}(z),$$
$$p'(x,y,z,t) = \sum p^{(n)}(x,y,t)\psi^{(n)}(z), \tag{2}$$

where $(u',v')$ is the anomaly of velocity component and $p'$ is the geopotential anomaly, representing the displacement of gravity potential by both the density anomaly and the volume transport as follows,

$$p' = gh'(x,y,t) + \frac{g}{\rho_0}\int_z^0 \rho'(x,y,z,t)\mathrm{d}z, \tag{3}$$

where $h'$ is the sea level anomaly and $\rho'$ is the density anomaly. To decompose the wave, following Toyoda et al. (2021), we integrate both sides of Eq. (2) in the vertical direction after multiplying $\psi^{(n)}$. Utilizing the orthogonality of the eigen function, we have

$$u^{(n)} = \left[ \int_{-H_b}^{0} u'\psi^{(n)}\mathrm{d}z \right] \left[ \int_{-H_b}^{0} (\psi^{(n)})^2\mathrm{d}z \right]^{-1},$$

$$v^{(n)} = \left[ \int_{-H_b}^{0} v'\psi^{(n)}\mathrm{d}z \right] \left[ \int_{-H_b}^{0} (\psi^{(n)})^2\mathrm{d}z \right]^{-1},$$

$$p^{(n)} = \left[ \int_{-H_b}^{0} p'\psi^{(n)}\mathrm{d}z \right] \left[ \int_{-H_b}^{0} (\psi^{(n)})^2\mathrm{d}z \right]^{-1}. \tag{4}$$

Thus, by substituting the $(u', v')$ and $p'$ obtained from the daily reanalysis data and the $\psi^{(n)}$ solved from Eq (1) to Eq (4), wave-induced anomalies in the corresponding baroclinic modes are extracted. In this study, $c^{(n)}$ and $\psi^{(n)}$ are solved using $N$ estimated by the mean vertical profiles of temperature and salinity data from 1993 to 2020 of the monthly reanalysis product. Although the yielded $\psi^{(n)}$ and $c^{(n)}$ are spatially varying (see Figure 1 and Figure 2), the $\psi^{(n)}$ in the above decomposition process is decoupled from the horizontal motions, given that the wave energy exchange between baroclinic modes is depressed during the wave propagation. Thus, the $\boldsymbol{V}^{(n)} = (u^{(n)}, v^{(n)})$ and $p^{(n)}$ are conserved in each mode, which makes the application of AGC scheme possible to detect equatorial waves by diagnosing the wave energy flux in the corresponding mode. Here we utilize the AGC level2 scheme by Aiki et al. (2017) as follows,

$$\boldsymbol{c_g}\overline{E} \approx \overline{\boldsymbol{V}^{(n)}p^{(n)}} + \nabla \times \overline{(p^{(n)}(\varphi^{(n)})}/2)\boldsymbol{z}, \tag{5}$$

where $E^{(n)} = \frac{1}{2}[(u^{(n)})^2 + (v^{(n)})^2 + (p^{(n)}/c^{(n)})^2]$ is the sum of kinetic and gravitational energies and $\boldsymbol{c_g}$ is the group velocity. The overline symbol in Eq. (5) is a phase average operator provided by Aiki et al. (2017). $\nabla \times \overline{(p^{(n)}(\varphi^{(n)})}/2)\boldsymbol{z}$ is an offset term, which includes a scalar quantity $\varphi^{(n)}$ solved by

$$\Delta\varphi^{(n)} - (f/c^{(n)})^2\varphi^{(n)} = q^{(n)}, \tag{6}$$

where $q^{(n)} = \frac{\partial v^{(n)}}{\partial x} - \frac{\partial u^{(n)}}{\partial y} - \frac{p^{(n)}f}{(c^{(n)})^2}$ is the Ertel's potential vorticity (EPV). Then, through Eq. (5), we can obtain the flux that orients the direction of the group velocity. The equatorial waves including KW, RW and even the mixed RW (Yanai wave) in the event year 2019 can therefore be detected from the reanalysis data by analysing the wave energy flux.

Equatorial waves are subject to linear shallow water equation (Matsuno, 1966), which are mainly forced by atmospheric forcing and some baroclinic instability. Due to the resonance mechanism, we can expect that waves will be more easily generated by forcing with the period that is close to the basin-mode period (defined as the four times of the period for KW to travel through the whole equatorial basin) (Cane and Moore, 1981; Brandt et al., 2016). Here we will focus on the first four gravest baroclinic modes, whose basin-mode periods (based on the average gravity wave speed) are no longer than 1.5 years, to analyse the wave-induced anomaly in the event year 2019. It should be pointed out here that, as waves in each baroclinic mode are already decoupled by Eq. (4), the selection of modes for the decomposition will not affect the decomposition result.

## 3 Results

### 3.1 Wave-induced variability

The extracted wave signal from the climatological data agrees well with previous results from linear model that are driven by the climatological forcing (Song and Aiki, 2020). The obtained variability of $u^{(n)}$ and $p^{(n)}$ at the equator is revealed largely in annual and semi-annual periods (see Figure 3). In the western basin, annual signals of both zonal velocity and geopotential are notable for the four modes. Positive zonal velocity anomalies appear in boreal spring and change to negative in the fall; correspondingly geopotential anomalies vary from negative to positive. However, from around $15°$W, the signals become nearly semi-annual (see Figure 3). In the central and eastern basin, for the second and third mode, the zonal velocity anomaly changes to negative from the summer rather than the fall and back to positive in the winter (see Figure 3 c & e); the geopotential anomaly appears to be positive in the spring and then oscillates in a period of around 0.7 year (see Figure 3 d & f). The disagreement between the western and eastern basin should be owing to the mixture of two waveguides: one originated from the central basin in May and the other from western basin in Sep. (Song and Aiki, 2020; Ding et al., 2009). The magnitude of wave signals in each mode is also significantly influenced by the basin-mode period. Specifically, the annual or semi-annual variability of both geopotential and zonal velocity becomes stronger when the period calculated by the averaged wave speed is closer to one or half a year (Song and Aiki, 2020; Claus et al., 2016). Thus the zonal velocity and geopotential anomalies in the second, third and fourth mode are all found prominent in Figure 3. However, the first-mode geopotential anomaly is unexpectedly prominent (Figure 3 b), which should be owing to the presence of thermal-induced seasonal variability in sea temperature and salinity.

We further investigate the wave-induced anomaly in 2019 to determine wave signal and its propagation during the event. Figure 4 confirms that, after removing the climatological values, the decomposed geopotential anomalies for the first four modes are able to reproduce the thermocline deepening for the 2019 winter event which is around one month ahead of the SST rising. The extracted waves in 2019 have demonstrated prominent variability of meridional velocity in all the presented modes with sign-alternating distributions along the equator (see Figure 5). It may suggest that mixed RWs featured by the cross-equatorial meridional velocity are excited by subseasonal forcing. The subseasonal forcing can as well excite KWs or gravity waves to enhance the variability of zonal velocity especially in the first mode due to its faster gravity wave speed (see Figure 5 a). As a result, the travelling signals of zonal velocity (see Figure 5) are prominent in all the four modes but the second and the fourth modes do not have notable propagation patterns as the climatological scenario. Correspondingly, Figure 5 b, d, f & h have suggested that from the fall of 2019, there are positive geopotential anomalies in all the four modes jointly causing the deepening of the thermocline. Those facts may indicate the additional KW trains induced by the subseasonal forcing that causes the negative displacement of thermocline thereby triggering the Atlantic Niño in 2019.

### 3.2 Horizontal wave energy flux

As the group velocity can be used to identify KWs and RWs, we first investigate the horizontal distribution of annual mean zonal wave flux (Figure 6 and Figure 7), which can effectively indicate the dominant location and type of waves in each baroclinic mode. In the climatological scenario (Figure 6), the AGC level2 scheme manifests clear equatorial KW trains originated from

the western Atlantic basin transporting the energy to the eastern boundary where it excites the reflected RWs and bring the energy back to the west. With the decrease of Rossby deformation radius, wave energy in high baroclinic modes tends to limit in a narrow latitude range. Thus we found a broader latitude coverage of westward energy flux in lower modes which may suggest the possible off-equatorial RWs (e.g. the westward energy flux within $15°$ in the north for the first mode and within $10°$ for the second mode). There is eastward energy flux that originates from the western boundary almost passing through the whole basin in the first mode (see Figure 6 a). This eastward energy flux in the eastern basin and its connection with the flux along the western boundary have not been seen in the research by Song and Aiki (2020) with linear ocean models. From Figure 6 a, the flux is likely due to the along-shore waves off the western boundary that bring the energy to the equatorial Atlantic from the extra-tropical region. Indeed, the planetary basin mode leads to the wave cycle that the coastally-trapped KW from high latitude will travel back to the equatorial region (Yang and Liu, 2003). Nevertheless this coastally-trapped KW along the western boundary can not be well reproduced in linear models because they usually apply radiant boundaries in both north and south rather than using enclosed basins as the real Atlantic Ocean (Matsuno, 1966; Song and Aiki, 2020). In the event year 2019, the annual mean wave flux for the subseasonal waves at the equator to transport the energy eastward is also prominent. In the second and third baroclinic modes, both the westward wave flux in the off-equatorial region and the eastward wave flux in the western basin exhibit greater strength than the climatological values in Figure 6. This result may suggest that the onset of the Atlantic Niño is associated with strong downwelling KWs induced by both subseasonal local forcing and off-equatorial energy.

We then investigate the evolution of wave energy flux at the equator in 2019. In Figure 8, by comparing the theoretical group velocity of KWs and RWs (solid red and blue) in corresponding modes, subseasonal waveguides can be detected. There are multiple KW trains passing through the basin in 2019. For all the four modes, strong KWs are excited in spring (from around Apr.) and fall (from around Sep.) keeping to transfer the energy for several month. Most of those KWs are originated from the western basin but some of them are notably strengthened in the central and eastern basin (e.g. in around Dec. for the first mode as Figure 8 a and for the third mode as Figure 8 c). However, KW trains do not guarantee the deepening of thermocline, the superposition of out-of-phase geopotential by waves in multiple modes may eliminate the displacement (Song et al., 2023a). In the fall and winter, we do find the positive geopotential anomaly (representing the deepening of thermocline to induce the positive SST anomaly) in the first three modes (see Figure 5). In the spring and summer, nevertheless, the second and fourth mode are dominated by the negative geopotential anomaly which likely eliminates the positive anomaly in the first and third mode so as to prevent the occurrence of the event in this season (see Figure 5). Also, by comparing the zonal wind stress anomaly (contours in Figure 8) with the waveguide, although most KWs have shown strong associations with westerly wind anomaly, there are still several mismatches between the local wind and the KW. For example, the summer KW train in the second mode, when it is excited, the western equatorial Atlantic is dominated by easterly wind anomaly. It may suggest that the KWs in the four modes with different phases of geopotential anomaly are not all excited by the local wind. Other energy sources should be taken into account, e.g. the westward energy flux in Figure 7 are indicating the possibility of reflected KWs to be excited by RWs. Multiple energy sources may cause the diversity of geopotential phase. However in Figure 8, the RW waveguide is difficult to be identified in Figure 8. low-frequency RWs (normally annual or interannual) are likely to be

obscured by subseasonal KW trains, since the local wave energy flux is calculated by combining the passing waves. Indeed, in the climatological scenario, RW trains are also prominent and can be easily detected (not shown).

### 3.3 Wave energy transfer process

In this section, we integrate the energy flux over several transections to further investigate the wave energy transfer process in 2019. As shown by the solid yellow lines in Figure 7, we have selected three sections (S2-S4) in the equatorial region
with the length of six degree (3°S-3°N) to match the recognized boundary of western Atlantic basin (40°W-20°W) and the ATL3 (20°W-0°), an off-equatorial section (S1) ($45°W$,4°N-10°N) to capture the westward energy flux (see Figure 7) that may approach the western boundary and transfer energy back to the equatorial region, and a meridional section S5 to check the down-wave influence on the region off the African coastline (see Figure 7).

The variances of the wave energy flux passing through the four selected transections in 2019 are presented in Figure 9.
Although the prominence of the eastward wave energy flux has been illustrated in all the four baroclinic modes at the onset stage of the SST event from around Oct. (see Figure 9), it is noteworthy that there is diversity among modes in transections with the largest passing energy flux. The third and fourth modes are both found to have the largest eastward energy flux to pass S4 but the energy flux passing S1, S2 and S3 are all weak. This may suggest that the waves in the two modes are locally forced in the ATL3 region. The energy source is located between S3 and S4 to excite the KW tranferring the energy east
through S4 (see solid red line in Figure 9 c & d). Meanwhile it may also excite RW to westward pass S3 (see solid blue line in Figure 9 c). Correspondingly, the eastward energy flux passing S3 for both the first and the second modes is found to be strong, which manifests that KWs are forced in the west of the ATL3. Moreover, in the second mode, the eastward energy flux peaks in around Oct. on S2 just after strong westward energy flux passing the off-equatorial transection S1 from Jun. to Oct., which may suggest a wave energy transfer route that sequentially passes S1, S2 and S3 to influence the ATL3 region. It hence illustrates
the influence of the wave energy from off-equatorial regions on the Atlantic Niño in 2019 to some extent. Additionally, we have only found prominent meridional fluxes in the first and third mode to pass S5. The peaks of meridional fluxes passing S5 occur almost simultaneously with the peaks of zonal fluxes passing S4, suggesting that the possible energy transferring from the equatorial region to the coastal region may mainly originate from the wind forcing in the eastern basin. Hence on the one hand, the wave exciting the coastally-trapped KWs is not remotely forced; on the other hand, the third-mode wave that carries
the highest energy to the coastal region (Figure 9 c) will dissipate rapidly when traveling off the equator due to its short Rossby deformation radius. The results therefore provide evidence that the anomalous SST event of the Benguela/Angola area in 2019 is mainly triggered by local forcing (Koungue et al., 2021).

We then investigate the spatial distribution of mean wind stress anomaly and wave energy flux in the Sep., Oct., and Nov. (Figure 10). The figure suggests that the peak of westerly wind anomaly during this event season is located at around 20°W
(see contours in Figure 10). The waveguide is revealed to have diverse origins in each mode (see color shadings in Figure 10). The seasonal mean wave energy fluxes in the second and third modes (Figure 10 b & c) have shown their discrepancies with the annual mean in Figure 7 b & c. In the second mode, the eastward energy flux by KWs originates from the western boundary so they may not be excited by the local wind in this season (Figure 10 b). Additionally, westward energy flux by reflected RWs

is found in the eastern basin suggesting that a strong KW is excited in summer for this mode (in agreement with Fig. 9b). For the third mode, however, it is revealed that the KW waveguide originates from the central basin (around $15°W$, close to the wind anomaly peak) and dominates the eastern basin (Figure 10c), which is in agreement with Figure 9c, confirming the high association between the third-mode KWs with local forcing.

## 4  Summary

This study decomposed the wave signal from the ocean reanalysis dataset and further diagnosed equatorial waves in the tropical Atlantic by applying a recently developed wave energy flux scheme. The equatorial waves for both climatological values and anomalies in 2019 are detected. Strong subseasonal equatorial KW trains were revealed in 2019, which we believed to associate with the onset of the SST event. All the first four modes are providing non-negligible contributions on the KW, but only the waves in the first three modes are causing prominent displacement of thermocline (see Figure 4). The discrepancy of energy transfer processes among the modes are notable. The third and fourth mode is mainly excited locally in the ATL3 region during the event while the wave energy of the first and second mode come from the west, likely to be affected by RWs through reflections. The source and contribution of the second-mode wave agrees with the result by Richter et al. (2022). In their analysis, the sea level anomaly (SLA) as well as the wind curl in the north tropical Atlantic has been diagnosed to reveal the propagation of RWs. Here, by wave energy flux, this study provides the evidence for the possible off-equatorial RWs and their influence on the Atlantic Niño event in 2019. Furthermore, in terms of the wave propagation, owing to the relatively slow group velocity, the wave flux of the higher mode (the third and fourth mode) is found to peak in the late Nov. and early Dec. holding the SST anomaly until the next Jan..

The extracted equatorial waves from both the climatological and the 2019 reanalysis data are demonstrating strong westward energy flux in the off-equatorial region especially in the first and second modes (seeing Figure 9 a & b and Figure 6 a & b) due to their relatively large Rossby deformation radius, which leads the wave energy to approach the western boundary for causing reflected KWs and eventually back to the equatorial region directly or along the coastline. The ray tracing by Wentzel-Kramers-Brillouin (WKB) approximation had revealed that reflected RW will transport their energy back to the equatorial region in a beta plane (Philander, 1978; Schopf et al., 1981; Claus et al., 2014). By means of linear ocean model with real coastline of the Atlantic basin, it has been demonstrated that the RW can be affected by the irregular African coastline so as to transport the energy back to the eastern Atlantic basin. However those models often missed the along-shore energy fluxes off the western boundary that also goes back to the equatorial region (Kopte et al., 2017; Song and Aiki, 2020; Kopte et al., 2018). Although previous research based on reanalysis dataset or ocean general circulation models (OGCM) can find general links between off-equatorial RWs and equatorial KWs by the propagation of SLA or the advection of SST (Richter et al., 2022; Foltz and McPhaden, 2010), still no concrete evidence was found for the baroclinic waves and their connections. This study, nevertheless, diagnosed the wave energy transfer route with reanalysis data so as to avoid the information loss with simple linear models and illustrate the wave energy transfer process in each baroclinic modes between the equatorial and off-equatorial regions.

In the proposal of using wave propagation to predict anomolous SST events in the equatorial Atlantic or the down-wave African coastal regions (Imbol Koungue et al., 2017, 2019; Koungue et al., 2021; Bachèlery et al., 2020), the diagnosis for equatorial waves in each mode is definitely the essential technique. However, SLA might be the only accessible observation with broad coverage for detecting wave signals, which is unable to illustrate the wave propagation in multiple vertical modes. The implement of ocean linear models and AGC wave energy flux scheme can yield group-velocity-based waveguide in each mode separately and better illustrate the dynamic feature of waves (Song and Aiki, 2020, 2021; Song et al., 2023a), but its deficiency is also obvious: only constant wave speed is allowed in the linear ocean model and the results crucially depend on the projection of wind anomaly into the corresponding mode (Claus et al., 2014; Brandt et al., 2011). This study attempted to employ the reanalysis dataset in the AGC scheme hence contributed to freeing the diagnosis of waveguide from the ocean linear model and promoting the waveguide-based warning system for Niño events. Still the presented diagnosis scheme has drawbacks in its utilization of assumed orthogonality for linear waves in cases where nonlinearity is relatively crucial. As a result, the decomposed variability can not correctly represent the contributions of each baroclinic modes. Indeed other attempts to decompose wave-induced variability e.g. projecting the velocity and geopotentionl to eigen functions by means of the least-square method applied by Tuchen et al. (2018) may also have similar problems. Nevertheless, it is expected that the error due to nonlinearity should be controllable in the linear-dominated equatorial basins. Certainly, the reliability of reanalysis dataset also limit the scheme usage. For instance, Tuchen et al. (2022) presented a multidecadal intensification in intraseasonal variability of velocity that is underestimated by altimetry data in the equatorial Atlantic. The underestimation may increase the uncertainty of using reanalysis data to extract correct wave signals, as most reanalysis dataset are relying on the correction by those satellite observations. However, as the equatorial waves had been revealed to have potentials in the prediction for the Atlantic Niños as well as the Benguela Niños in the down-wave region (Imbol Koungue et al., 2017; Bachèlery et al., 2020; Richter et al., 2022; Song et al., 2023b), we expect the rapid wave diagnosis using reanalysis data can provide a useful tool in the associated community.

*Data availability.* The temperature, salinity and velocity data of GLORYS12V1 product is from (https://data.marine.copernicus.eu/product/GLOBAL_MULTIYEAR_PHY_001_030/services). The OISST data used to validate the model is available via (http://apdrc.soest.hawaii.edu:80/dods/public_data/NOAA_SST/OISST/monthly). The wind data of ERA5 can be accessed via (http://apdrc.soest.hawaii.edu:80/dods/public_data/Reanalysis_Data/ERA5/monthly_2d/Surface).

*Author contributions.* Qingyang Song and Hidenori Aiki conceived the study. Qingyang Song processed the reanalysis data and performed the wave energy analysis with scientific insight of Hidenori Aiki. The paper was written by Qingyang Song with contributions of the co-author.

*Competing interests.* The authors declare that they have no conflicts of interest.

*Acknowledgements.* The authors gratefully acknowledge the financial support from the Young Scientists Fund of the National Natural Science Foundation of China (Grant Numbers 42206008) and the China Postdoctoral Science Foundation (Grant Numbers 2021M701040).

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

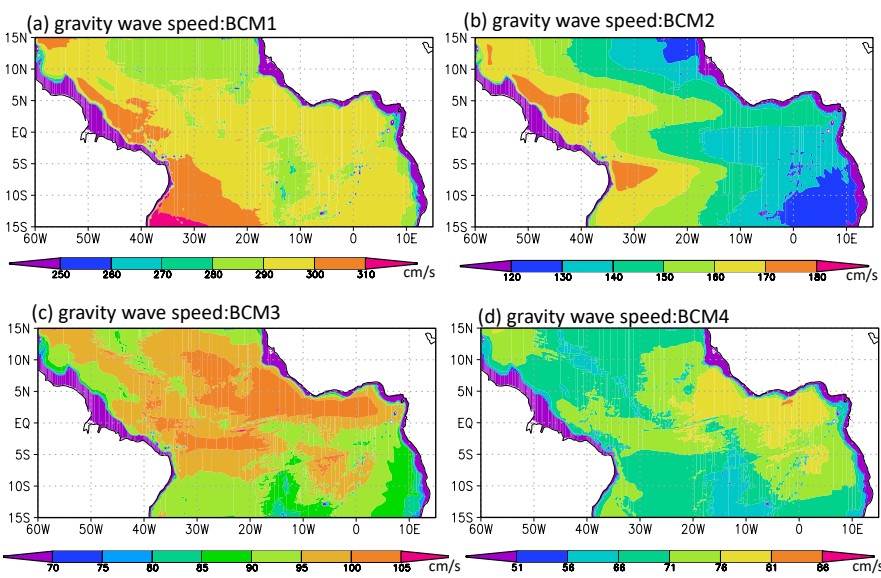

**Figure 1.** Gravity wave speed in the tropical Atlantic for the (a) first, (b) second, (c) third and (d) fourth baroclinic mode (BCM). The gravity wave speed is solved by Eq. (1) with mean vertical profile of salinity and temperature from GLORYS12V1 product over 1993 to 2020.

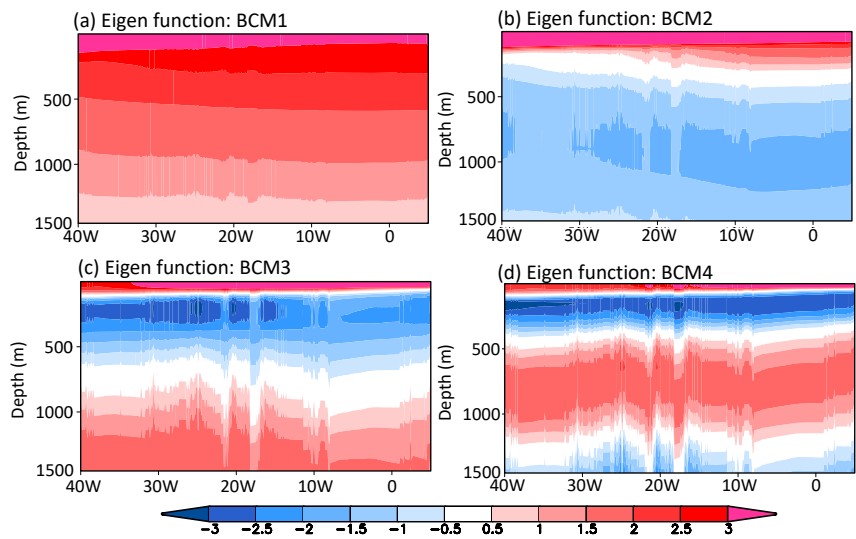

**Figure 2.** Eigen functions at the equator for the (a) first, (b) second, (c) third and (d) fourth baroclinic mode (BCM). The eigen function is also solved by by Eq. (1) with mean vertical profile of salinity and temperature from GLORYS12V1 product over 1993 to 2020, which corresponds to the gravity wave speed (eigen value) in Figure 1.

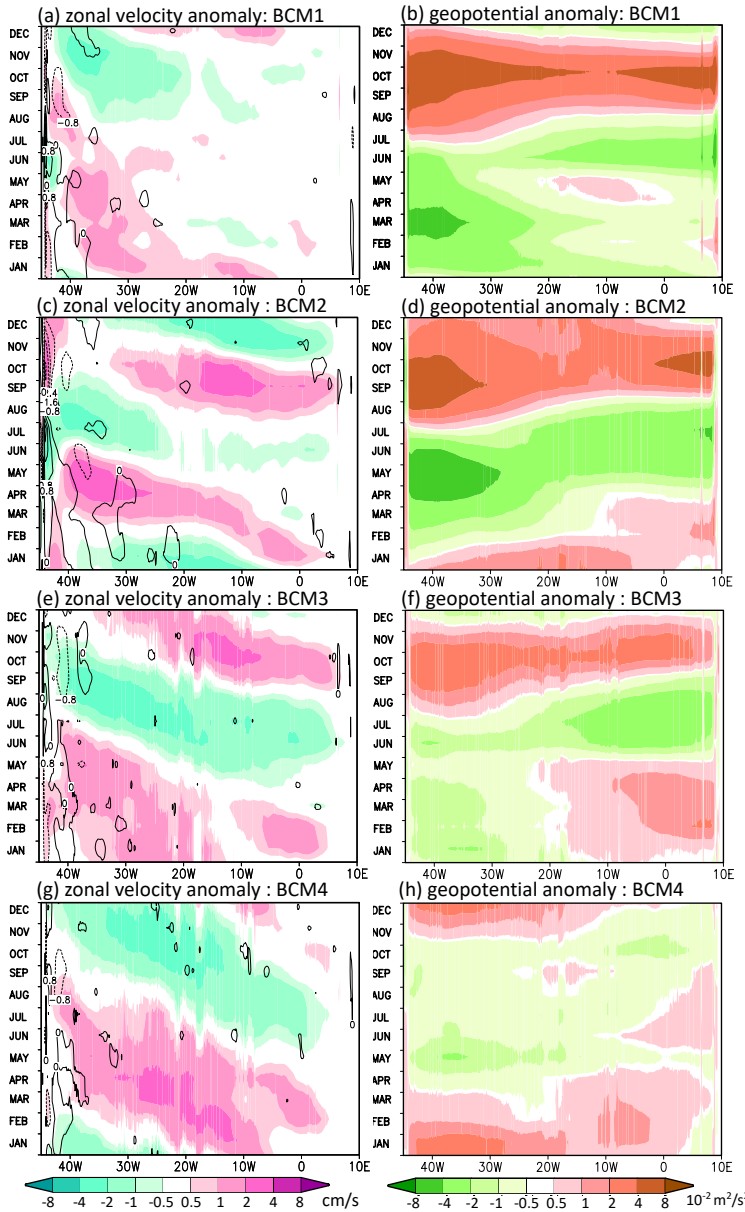

**Figure 3.** Hovmöller diagram for the climatology of velocity (left panels) and geopotential (right panels) at the equator decomposed from the daily GLORYS12V1 product in the first (a,b), second (c,d), third (e,f) and fourth (g,h) baroclinic modes (BCM). Color shadings in the left panels are zonal velocity $u^{(n)}$; contours in the left panels are meridional velocity $v^{(n)}$ with the interval of 0.8 cm/s. Color shadings in the right panels are geopotential $p^{(n)}$.

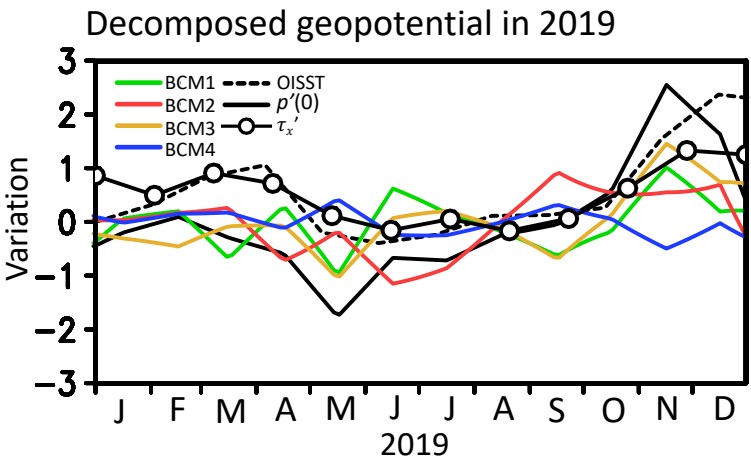

**Figure 4.** Timeseries of geopotential anomaly at the sea surface $p'(0)$ (solid black line), SST anomaly (dashed black line) from OI-SST (Optimum Interpolation Sea Surface Temperature) dataset, zonal wind stress anomaly (circle-marked line) from ERA5 (European Centre for Medium-Range Weather Forecasts Reanalysis v5) dataset and the decomposed geopotential for the first (BCM1, solid green line), second (BCM2, solid red line), third (BCM3, solid yellow line) and fourth (BCM4, solid blue line) baroclinic mode in 2019. The $p'(0)$ is calculated following Eq. (4) with the decomposed geopotential $p^{(n)}\psi^{(n)}(0)$. The anomalies of SST and geopotential are averaged over the Atlantic 3 region (ATL3, $3°$S-$3°$N,$20°$W-$0°$); the wind stress anomaly is averaged over the western Atlantic basin ($3°$S-$3°$N,$40°$W-$20°$). The anomalies are normalized by the variance of SST, zonal wind stress and $p'(0)$ from 1992 to 2019 respectively.

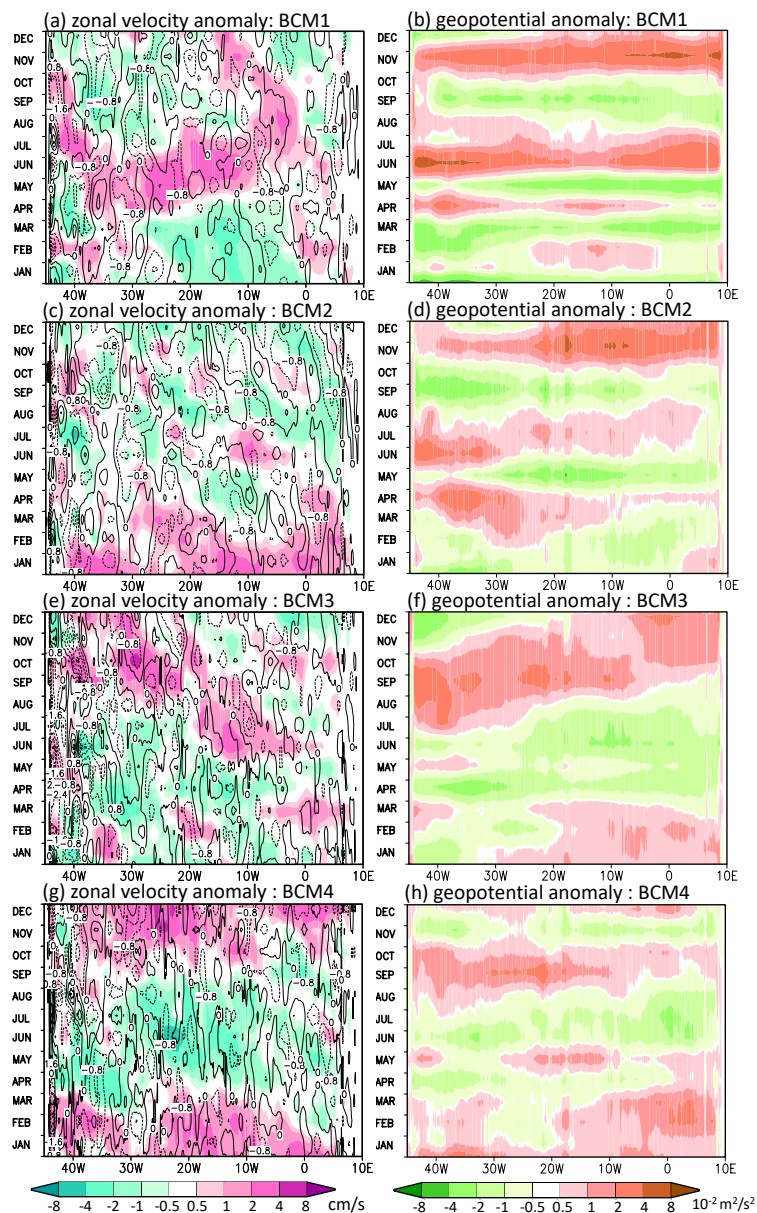

**Figure 5.** Same as Figure 3 but for the anomaly in the event year 2019.

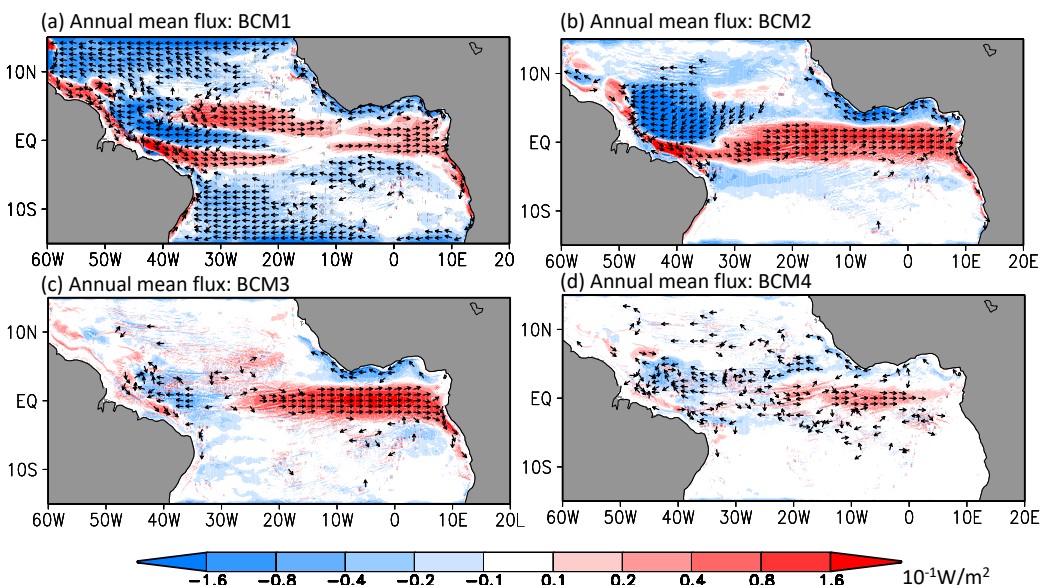

**Figure 6.** Annual mean climatological AGC flux for the (a) first, (b) second, (c) third and (d) fourth baroclinic modes (BCM). Color shadings are the zonal component of the AGC flux; arrows with a constant length indicate directions of flux vectors.

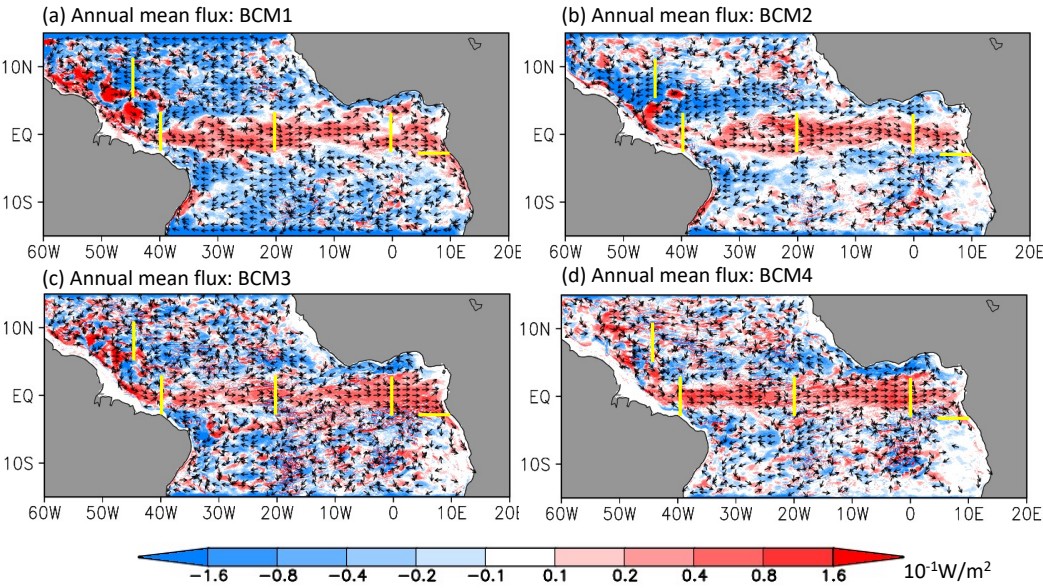

**Figure 7.** Same as Figure 6 but for the anomaly in 2019. Solid yellow lines indicate four meridional and one zonal transections: S1 (4°N-10°N, 45°W), S2 (3°S-3°N, 40°W), S3 (3°S-3°N, 20°W,), S4 (3°S-3°N, 0°) and S5 (3°S, 5°E-10°E).

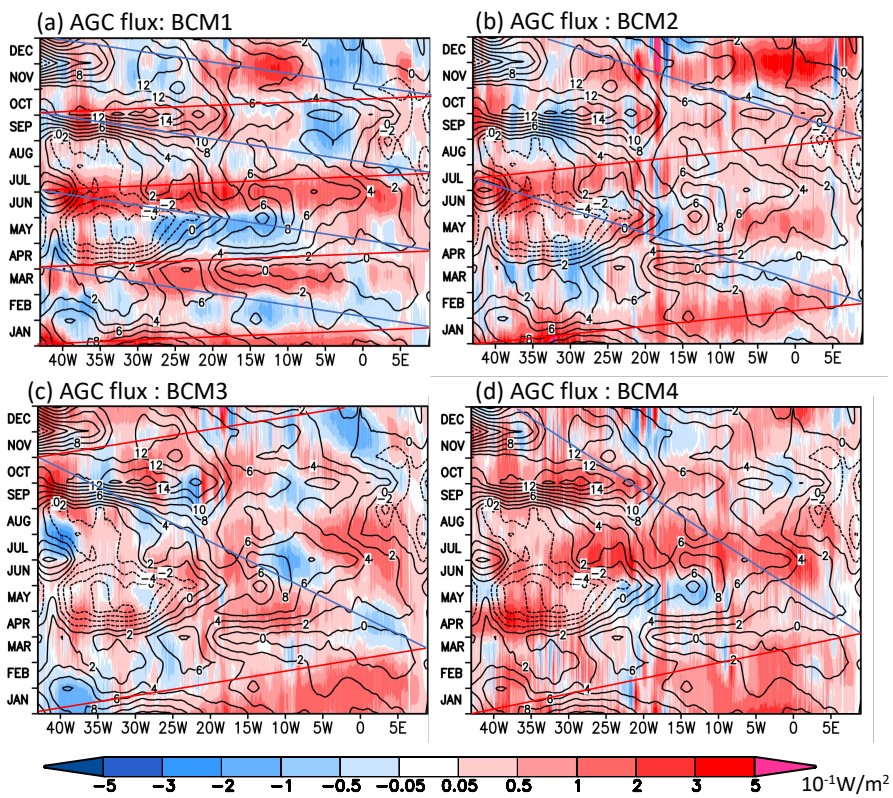

**Figure 8.** Hovmöller diagram for the AGC flux in 2019 at the equator in the (a) first, (b) second, (c) third and (d) fourth baroclinic modes (BCM). Color shadings are the zonal AGC flux at the equator; contours are the anomaly of zonal wind stress with the interval of $2 \times 10^{-3} \mathrm{N/m}^2$. The solid red (blue) line represents averaged group velocity for the KW (RW) in the BCM1 of around 2.9 m/s (1.0 m/s), BCM2 of around 1.4 m/s (0.5 m/s), BCM3 of around 0.9 m/s (0.3 m/s) and BCM4 of around 0.7 m/s (0.2 m/s) at the equator (see Figure 8).

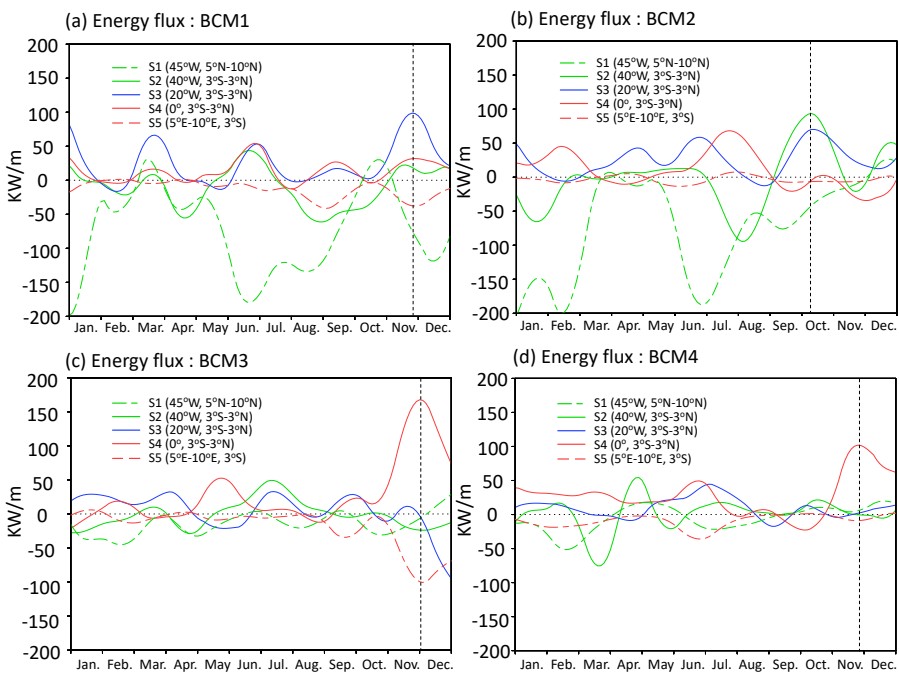

**Figure 9.** Timeseries of AGC flux in 2019 for the (a) first, (b) second, (c) third and (d) fourth baroclinic modes (BCM) passing the zonal transection S1 (dashed green line), S2 (solid green line), S3 (solid blue line) and S4 (solid red line) and meridional section S5 (dashed red line). Dotted black lines indicate the peaks of the positive flux passing the sections in 2019.

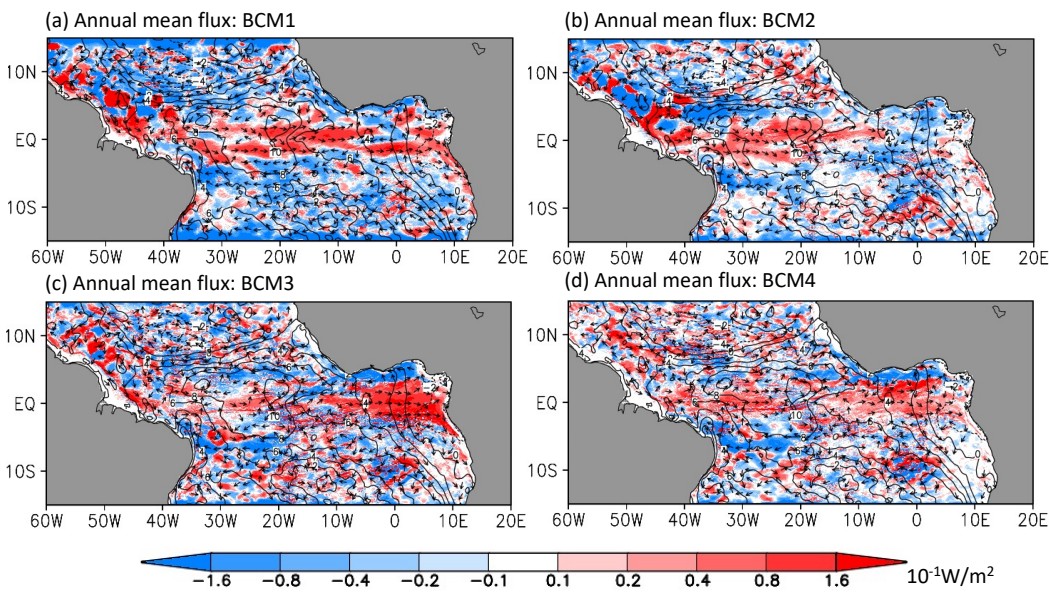

**Figure 10.** Seasonal mean AGC flux for the (a) first, (b) second, (c) third and (d) fourth baroclinic modes (BCM) over September, October and November. Color shadings are the zonal component of the AGC flux; contours are the anomaly of zonal wind stress with the interval of $2 \times 10^{-3} \mathrm{N/m^2}$ averaged over the same season; arrows with a constant length indicate the directions of the flux vector.