# Peer review of "Equatorial wave diagnosis for the Atlantic Niño in 2019 with an ocean reanalysis."

_EGUsphere, 2023_

## Author Comment (AC1)

**Response to Referee 1**

**Summary**

**Using a newly developed wave flux diagnostic as a tool, the authors examine the genesis of the eastern equatorial Atlantic warming in 2019. The model allows analyzing the contributions of individual baroclinic modes, and the authors examine the first four of those. They find that the 3rd and 4th modes have substantial contributions to the warming and that these modes are locally forced. Prior those higher modes, there is a 2nd mode Rossby wave that appears to be excited in the off-equatorial region and reflected into a Kelvin wave at the western boundary. The authors suggest that this 2nd mode Kelvin wave helped to precondition the event.**

**The authors show some interesting results and I believe the diagnostic tool could be useful for obtaining a deeper understanding of equatorial Atlantic variability. It is less clear, however, how useful this diagnostic tool is for prediction purposes and for quantifying individual contributions. Furthermore, the English in the manuscript could use some editing. Detailed comments follow.**

**Major Comments**

**1) Figure 8 shows some good evidence for Kelvin wave propagation. For the Rossby waves, however, there is little agreement with the theoretical phase speed and, in fact, the data shows little evidence for any westward propagation. Does this mean that the Kelvin waves are mostly transmitted into coastally trapped waves at the eastern boundary? Do you have a way of quantifying the relative amounts of reflected and transmitted energy?**

Thank you for the comment. The local wave energy flux at one grid is essentially determined by averaging the flux from both Kelvin and Rossby waves. Indeed, in climatological scenarios, the waveguide of Rossby wave is clear and agrees well with the theoretical group velocity (see the Figure below). However in Figure 8, when the climatological variability is excluded and high-frequency (subseasonal) Kelvin waves dominate, the low-frequential Rossby waveguide is obscured by subseasonal Kelvin waves. Although high-frequency wave signal is not likely to excite reflected Rossby waves, it can be transmitted to inertia oscillations and dissipates rapidly. From only Figure 8, it is  hard to say how much energy is transferred into coastally trapped waves.  To give a rough estimation of the energy entering the off-equatorial area, we have calculated the meridional flux in the newly selected transection at 3ºS (see Figure 9 in the revised manuscript). It turns out  that in the 2019 event, waves only in the first and third mode will transport significant energy to the coastal region.

[Figure]

XT diagram for AGC flux at the equator. Same as Figure 8 in the revised manuscript but for the climatological wave signal.

**2) How well do the Kelvin waves correspond to the surface wind stress forcing in terms of location and timing? Is there a proportionality between the strength of the equatorial wind stress anomalies and the excited wave energy? This could also help to further clarify the relative contributions from off-equatorial and equatorial waves.**

Thank you for the comment. To reveal the association between surface wind and the wave energy, we have employed wind stress from the ERA5 dataset and added it in our figures. The updates of the figures regarding the wind information include: 1. In Figure 4, we added the timeseries of the zonal wind anomaly averaged in the western equatorial Atlantic (40ºW-20ºW, 3ºS-3ºN, which is traditionally regarded as the wave source region) to show the correlations between the geopotential anomaly and the wind anomaly; 2. In Figure 8, the XT diagram of zonal wind anomaly is drawn as contours to demonstrate its correspondence to the excitement of waves at the equator; 3. In Figure 10, we have drawn the horizontal distribution of mean wave energy flux and mean zonal wind anomaly to compare the source region of wave energy with the pattern of wind anomaly in the event season.

[Figure]

Figure 4 in the revised manuscript.  Solid black line is the zonal wind stress averaged in the western equatorial basin.

[Figure]

Figure 8 in the revised manuscript. Contours are the zonal wind stress anomaly with the interval of 0.002 N/m2.

[Figure]

Figure 10 in the revised manuscript. Contours are the zonal wind stress anomaly with the interval of 0.002 N/m2.

Those figures have suggested the strong association between locally equatorial forcing with high-mode waves (in the third and fourth mode shown as Figure 8 and 10). Meanwhile, the mismatch between the wind anomaly peak and the wave energy source in Figure 10b confirms the contribution of off-equatorial waves to the equatorial wave energy in the second mode. The detailed description and analysis for those updated figures have also been given in the revised manuscript (line 172-175 and line 208-217).

**3) The 3rd and 4th modes seem to make a strong contribution to the equatorial Atlantic warm event. Since these waves seem to be excited locally, they offer very little predictive potential. Does this diminish the prospect of predicting similar events? Can you estimate the relative contributions from the 2nd (reflected) mode and the 3rd and 4th modes?**

Thank you for the comment. There are several metrics that can help to evaluate the contribution of waves in each mode. For example, using wave energy E in Eq. (5) to directly calculate the wave energy. However, Song et. al (2023) has indicated that the square fashion of potential energy applied in either Eq. (5) or other research such as (Imbol Koungue et al., 2017) can not reflect the influence of linear superposition by waves in multiple vertical modes on the thermocline displacement. Showing the geopotential anomaly with the sign should be more proper. Thus, by giving Figure 4, we think the relative contribution of each mode on the event has already been revealed, that the 1st-3rd modes are of similar importance, while the wave in the 4th mode makes the negative contribution on the thermocline deepening. We agree that the origin of the Kelvin waveguide in the 3rd mode is located at around 15oW (as shown in Fig. 8 and Fig. 10c), taking around 1.5 months to pass the ATL3 region based on its wave speed (around 0.9 m/s), which will crucially shorten the leading time for skillful prediction. However it also takes one or two months for the displacement of thermocline to affect SST so that the locally-forced 3rd-mode wave may still have some potential. Also, the high-mode wave can make contributions to the excitement of the associated coastally-trapped Kelvin waves so as to support the prediction for SST events in the down-wave offshore region along the African coastline. On the other hand, the diagnosis of waveguide should be the previous step for the

designing of predicting skills, in which sense the revelation of locally-forced waves is also meaningful.

Song, Qingyang, Hidenori Aiki, and Youmin Tang. "The role of equatorially forced waves in triggering Benguela Niño/Niña as investigated by an energy flux diagnosis." *Journal of Geophysical Research: Oceans* (2023): e2022JC019272.

Imbol Koungue, Rodrigue Anicet, Serena Illig, and Mathieu Rouault. "Role of interannual K elvin wave propagations in the equatorial A tlantic on the A ngola B enguela C urrent system." *Journal of Geophysical Research: Oceans* 122.6 (2017): 4685-4703.

**4) The potential negative interference of baroclinic modes is an interesting argument. It would be interesting to examine this in more detail and to show how important this effect is and if it is a systematic feature. Previous studies have shown that similar (average) equatorial Atlantic wind forcing can have different outcomes in terms of the ATL3 SST (Richter et al. 2013; Martin-Rey et al. 2019). Could this be explained by wave interference? This also connects to the inconsistent influence of ENSO on the equatorial Atlantic (Chang et al. 2006).**

Thank you for the comment. We agree that the wave interference might be possible to cause the diversity of SST events. Song (2023 a,b) designed several numerical experiments with linear ocean models to investigate Atlantic/Benguela Niño events in recent 30 years. They found that linear superposition of out-phase waves among different modes can eliminate the displacement of thermocline so as to prevent the onset of SST events even if both the wind anomaly and wave signal in each mode are prominent. However, the non-linear interaction and energy transfers between modes are still missing in their studies owing to the deficiency of linear models. In this study, although the reanalysis data contain much richer information than the result by linear ocean models, the decomposition process however also discards the possible evidence of non-linear interactions. On the other hand, going deep into this discussion might be beyond the scope of this study to introduce the useful tool for equatorial wave diagnosis.

Song, Qingyang, Hidenori Aiki, and Youmin Tang. "The role of equatorially forced waves in triggering Benguela Niño/Niña as investigated by an energy flux diagnosis." *Journal of Geophysical Research: Oceans* (2023): e2022JC019272.

Song, Qingyang, Youmin Tang, and Hidenori Aiki. "Dual wave energy sources for the Atlantic Niño events identified by wave energy flux in case studies." *Journal of Geophysical Research: Oceans* (2023): e2023JC019972.

**5) How do your results compare with those of Richter et al. (2022) who specifically examined the genesis of the 2019 event?**

Thank you for the comment. Richter et al. (2022) have shown a possible link of the off-equatorial Rossby wave with the onset of the 2019 event. In their analysis, they diagnosed the sea level anomaly as well as the wind curl in the north tropical Atlantic to reveal the propagation of Rossby waves. This study agrees well with their results. By wave energy flux, this study

confirms the off-equatorial Rossby waves and their influence on the Atlantic Niño event in 2019 to some extent. Furthermore, we have found that the second mode is the most prominent mode to exert this off-equatorial influence on the event. In line (226-229), we have put a short discussion in the context of the study by Richter et al. (2022) to give our point.

**6) In Fig. 9, modes 3 and 4 seem to be maximum at the 0-meridian in late fall/early winter. This is the eastern edge of the ATL3 region and so it is not clear how important these two modes are for the equatorial warm event. Does the strong amplitude at this longitude indicate that the waves pass through to the eastern boundary? Is there evidence that they are transmitted into coastally trapped waves?**

Thank you for the comment. As we have explained in the comment 3, the respective contribution of waves in each vertical mode to the event can be roughly evaluated in Figure 4 of the revised manuscript, where the averaged geopotential is a good metric to represent the displacement of thermocline hence can reflect the influence of waves on SST. From Figure 4, the waves in mode 3 are definitely crucial for the SST event, while the wave in mode 4 is insignificant. This result agrees well with the numerical experiments using linear ocean models by Song et al (2023). Regarding the waves in the eastern basin, to investigate whether they are transmitted into coastally trapped waves, we have selected an extra meridional transection as S5 at 3oS off the coastline with the width of 5o and calculated the meridional flux passing this section (see the revised Figure 7 and Figure 9 below). We found that "*The peaks of meridional fluxes passing S5 occur almost simultaneously with the peaks of zonal fluxes passing S4, suggesting that the possible energy transferring from the equatorial region to the coastal region may mainly originate from the wind forcing in the eastern basin. Hence on the one hand, the wave exciting the coastally-trapped KWs is not remotely forced; on the other hand, the third-mode wave that carries the highest energy to the coastal region (Figure 9 c) will dissipate rapidly when traveling off the equator due to its short Rossby deformation radius. The results therefore provide evidence that the anomalous SST event of the Benguela/Angola area in 2019 is mainly triggered by local forcing*" (line 201-207 in the revised manuscript).

[Figure]

Figure 7 in the revised manuscript.

[Figure]

Figure 9 in the revised manuscript. The dashed red line is the newly selected meridional transection at 3ºS.

Song, Qingyang, Youmin Tang, and Hidenori Aiki. "Dual wave energy sources for the Atlantic Niño events identified by wave energy flux in case studies." *Journal of Geophysical Research: Oceans* (2023): e2023JC019972.

**7) In Fig. 9b, the authors stress the S1 peak in September and how it is followed by an opposite signed peak in S2. This, however, is only a secondary peak. A much stronger peak occurs in late June. How do they authors explain that this peak is not followed by a peak in S2?**

Thank you for the comment. The local flux in Figure 9 is determined by both the Kelvin and Rossby waves passing the transaction. Therefore, to explain the flux variation in S2, we should go back to Figure 8 to check the energy transfer route. In Figure 8b, it is found that Kelvin waves are holding from the boreal summer, meanwhile in late June, a strong Rossby wave is just approaching S2 bringing the negative (westward) flux to pass S2 in Figure 9b. In this sense, the reflected Kelvin waves by off-equatorial waves passing S1may indeed cause positive (eastward) flux in S2 from summer, however it is balanced by the negative (westward) flux of equatorial Rossby waves until Sep.. But we agree that the original sentence can cause confusion. Hence we have revised this sentence as "*Moreover, in the second mode, the eastward energy flux peaks in around Oct. on S2 just after strong westward energy flux passing the off-equatorial transection S1 from Jun. to Oct., which may suggest a wave energy transfer route that sequentially passes S1, S2 and S3 to influence the ATL3 region. It hence illustrates the influence of the wave energy from off-equatorial regions on the Atlantic Niño in 2019 to some extent.*" (line 197-200 of the revised manuscript).

**Minor Comments**

**1) l. 14: Please provide some references for the Atlantic Niño phenomenon.**

Thank you for the comment. We have put Giannini et al. (2003) as the reference for the Atlantic Niño phenomenon.

Giannini, Alessandra, R. Saravanan, and Ping Chang. "Oceanic forcing of Sahel rainfall on interannual to interdecadal time scales." *Science* 302.5647 (2003): 1027-1030.

**2) ll. 16-19: I believe Rodriguez-Fonseca et al. (2009) is an important paper for the influence of the Atlantic Niño on ENSO and should be referenced here.**

Thank you for the comment. We have added Rodriguez-Fonseca et al. (2009) as the reference for the remote influence of the Atlantic Niño.

Rodríguez-Fonseca, Belén, et al. "Are Atlantic Niños enhancing Pacific ENSO events in recent decades?." *Geophysical Research Letters* 36.20 (2009).

**3) Do you use realistic bottom topography to obtain Hb (depth of ocean bottom)?**

Thank you for the comment. Yes, although we did not employ independent topography data, as the reanalysis dataset of 50 vertical levels are fully involved, the realistic bottom topography is literally applied.

**4) In the methods section you say that you use long-term climatological temperature and salinity distributions to calculate the modes. Would your results change much if you used the actual temperature and salinity from 2019?**

Thank you for the comment. If only the annual mean temperature and salinity in 2019 is applied to calculate the modes, both the obtained eigen vector and eigen value (gravity wave speed) for each mode will be different so as to affect not only the total wave energy but also the contribution of each mode. However, when you apply only the TS data in 2019 to represent the mean state, you indeed improperly estimate the wave-induced anomaly and introduce errors, which is inappropriate in this study.

**5) There are many places were editing the English would improve the readability of the manuscript. A few examples are listed below:**

1.   **a) l. 41: "in addition to involve in-situ and altimetric data"**

**The meaning is not clear. Maybe "in addition to including in-situ and altimetric data" was meant?**

1.   **b) ll. 41-42: "The implementation of ocean linear model in those proposals are necessary"**

**Maybe "This will require the use of linear ocean models" was meant?**

1.   **c) l. 97: "Then though Eq. (5)" -. Then, through Eq. (5),"**
2.   **d) ll. 97-98: "pressure flux is redirected to the direction of the group velocity"**

**I am not quite sure I follow this. Please rephrase.**

1. **e) l. 99: "are hence able to be detected" -> "can therefore be detected"**
2. **f) ll. 126-127: "It may suggest that mixed RWs in the nature of cross equatorial meridional velocity are raised by subseasonal forcing."**

**"in the nature of": perhaps "in the form of" was meant?**

**"raised by" -> "excited by"**

Thank you for your careful review. All the mentioned errors have been revised and a proofreading has been made through the manuscript to make it better readerable.

**References**

Chang, P., Fang, Y., Saravanan, R. et al. The cause of the fragile relationship between the Pacific El Niño and the Atlantic Niño. Nature 443, 324–328 (2006). https://doi.org/10.1038/nature05053

Martín-Rey, M., Polo, I., Rodríguez-Fonseca, B., Lazar, A., & Losada, T. (2019). Ocean dynamics shapes the structure and timing of Atlantic Equatorial Modes. *Journal of Geophysical Research: Oceans*, 124, 7529– 7544. https://doi.org/10.1029/2019JC015030

Richter, I., Behera, S. K., Masumoto, Y., Taguchi, B., Sasaki, H., & Yamagata, T. (2013). Multiple causes of interannual sea surface temperature variability in the equatorial Atlantic Ocean. *Nature Geoscience*, 6, 43– 47.

Rodríguez-Fonseca, B., Polo, I., García-Serrano, J., Losada, T., Mohino, E., Mechoso, C. R., and Kucharski, F. (2009), Are Atlantic Niños enhancing Pacific ENSO events in recent decades? Geophys. Res. Lett., 36, L20705, doi:10.1029/2009GL040048.

---

## Author Comment (AC2)

**Response to referee 2**

**General comments:**

**In their study „Equatorial wave diagnosis for the Atlantic Nino in 2019 with an ocean reanalysis", the authors use output from the CMEMS GLORYS12V1 reanalysis product to which they apply a wave energy flux scheme to study equatorial wave propagation during the 2019 Atlantic Nino event. The authors find that both equatorial Kelvin waves (locally forced) and off-equatorial Rossby waves (reflecting into equatorial Kelvin waves at the western boundary) contributed to triggering the Atlantic Nino in late 2019. Their diagnostic tool allows for modal decomposition showing that third and fourth baroclinic mode Kelvin waves are locally forced, while the second baroclinic mode Kelvin wave was remotely forced by off-equatorial Rossby waves. The authors suggest to apply this wave energy flux scheme to real-time data in order to better predict Atlantic Nino events.**

**This study examines an important research topic and advocate (rightfully) that a more skillful prediction of extreme events like the 2019 Atlantic Ninos is needed. The results are interesting and potentially helpful for a better understanding of which waves are at play during Atlantic Nino. However, several questions remain. Most importantly, the study is missing a validation of the reanalysis product with observations. Further, even though the authors motivate their study with the need for a better prediction of Atlantic Nino, it remains unclear how the presented study might be helpful in doing so. Lastly, a more thorough discussion with recent studies would be helpful to highlight the new findings of this study.**

**I am listing my comments and suggestions below. Based on these, I recommend major revisions of the manuscript before publication.**

**Major comments:**

**A potential weakness of the presented analysis is the use of reanalysis data on the equator without providing any validation with observations. Most ocean reanalysis products have been found to underestimate observed velocity variability on the equator (see e.g., Tuchen et al., 2022a). How is GLORYS12V1 handling this issue? A comparison of equatorial velocity from GLORYS12V1 with observations from PIRATA buoys would be a meaningful assessment of the reanalysis' capability of reproducing realistic velocity signals in the tropical Atlantic Ocean. Potential data sets of velocity are provided at 0°, 23°W (Tuchen et al., 2022b), at 0°, 10°W (Brandt et al., 2021) or at 4°N, 23°W (Perez et al., 2019). Additional, but possibly shorter, timeseries are available at other PIRATA sites: https://www.pmel.noaa.gov/tao/drupal/disdel/.**

Thank you very much. We fully understand the referee's concern about the quality of reanalysis dataset. However, those assessments have already been done by the CMEMS team. In their paper and quality information documents (LelloucheJean-Michel 2021 ;Marie Drévillon et al. 2022), they have compared the GLORYS12 dataset with the in-situ observations including the PIRATA and TAO buoys. Both the surface velocity and velocity profiles are validated. The mean correlation between the reanalysis data and observations at (23oW, 0oE) over the whole

velocity profiles is around 0.6. In the mixed layer, the RMSE is around 0.1 m/s where the correlation exceeds 0.7. From the report by Marie Drévillon et al. 2022, we found that the GLORYS12 dataset does slightly underestimate the zonal velocity variability by around 5%-10% in the mixed layer from 15m-80m at (23oW, 0oE). Marie Drévillon et al. (2022) have described the assimilation scheme and listed the TS and SSH data assimilated to produce the GLORYS12 in detail. The velocity observations were only used for validation in the dataset. We will not go deep into the dataset validation which exceeds the scope of this study, instead we have given a short description about the quality of GLORYS12 in the Section 2 (line 67-72) and put the (Marie Drévillon et al. 2022) as the reference for readers to check.

Jean-Michel, Lellouche, et al. "The Copernicus global 1/12 oceanic and sea ice GLORYS12 reanalysis." Frontiers in Earth Science 9 (2021): 698876

Marie, D., Jean-Michel, L., Charly, R., Gilles, G., Clément, B., Olga, H., and Romain, B.-B. " Quality information document for Global Ocean Reanalysis Products GLOBAL_REANALYSIS_PHY_001_030." (2022)

**Lines 153-165: The comparison between theoretical Kelvin wave and Rossby wave propagation with the observed AGC flux (Fig. 8) is not very convincing. There is hardly any westward propagation visible in all four modes that would fit to theoretical Rossby wave propagation. There is better evidence for Kelvin wave propagation, but I think the authors have to clearly address and discuss this shortcoming which is not done in this paragraph.**
**The authors motivate their study by mentioning an "early warning system" that is needed for a better prediction of such extreme warm events. It would be meaningful if the authors pick up this motivation and further evaluate and discuss how their study is helping to achieve this goal. What is the potential on more skillful predictability of Atlantic Nino events when using the AGC scheme and real-time data or reanalysis output?**

Thank you for the comment. The local wave energy flux at one grid is determined by combining the flux from both Kelvin and Rossby waves. Thus, in climatological scenarios, the waveguide of Rossby wave is clear and agrees well with the theoretical group velocity (see the Figure below). However in Figure 8, when the climatological variability is excluded and high-frequential Kelvin waves dominate, the low-frequential Rossby waveguide is obscured by subseasonal Kelvin waves. We hence have clarified this in line 178-181 as "*The RW waveguide is difficult to be identified in Figure 8. low-frequency RWs (normally annual or interannual) are likely to be obscured by subseasonal KW trains, since the local wave energy flux is calculated by combining the passing waves. Indeed, in the climatological scenario, RW trains are also prominent and can be easily detected.*"

[Figure]

XT diagram for AGC flux at the equator. Same as Figure 8 in the revised manuscript but for the climatological wave signal.

Regarding the motivation, as we have mentioned in the introduction section, using the dynamics of equatorial waves to predict anomalous SST events was proposed by many studies (Imbol Koungue et al. 2017, 2019; Song et al. 2023 ). Wave energy can be transported from its origin to the concerned region in months following the group velocity of the corresponding vertical modes. Compared with the variation of geopotential and SLA, using wave energy flux to predict the event is more reasonable in dynamics and may have potentials for an extended leading time if the wave propagates from a remote region. Hence for the warning system, the essential technique is the diagnosis of waves in each mode. However even by using 1.5-layer ocean linear models to separately simulate equatorial waves in each mode, it is still difficult to diagnose waveguide. The deficiencies of linear ocean models are also obvious: only constant wave speed is allowed and their results crucially depend on the projection of wind anomaly into the corresponding mode. This study attempted to employ the reanalysis dataset in the AGC scheme hence contributed to freeing the diagnosis of waveguide from the ocean linear model so that the warning system based on equatorial waveguide could be promoted. We have added statements in the Summary section (line 246-255) for highlighting our motivations and contributions in the research context.

Imbol Koungue, Rodrigue Anicet, Serena Illig, and Mathieu Rouault. "Role of interannual K elvin wave propagations in the equatorial A tlantic on the A ngola B enguela C urrent system." *Journal of Geophysical Research: Oceans* 122.6 (2017): 4685-4703.

Imbol Koungue, Rodrigue Anicet, et al. "Benguela Niños and Benguela Niñas in forced ocean simulation from 1958 to 2015." *Journal of Geophysical Research: Oceans* 124.8 (2019): 5923-5951.

Song, Qingyang, Hidenori Aiki, and Youmin Tang. "The role of equatorially forced waves in triggering Benguela Niño/Niña as investigated by an energy flux diagnosis." *Journal of Geophysical Research: Oceans* (2023): e2022JC019272.

**Minor points:**

- **1. Analogously to velocity (major comment 1), how do estimates of vertical N profiles compare to observations? Errors in N would directly propagate into errors of gravity wave speed and y(n).**

Thank you for the comment. Indeed, the vertical N derived from the density profile is crucial for decomposition of the equatorial waves. Also, Marie Drévillon et al. (2022) has already evaluated the quality of TS data in the information documents of GLORYS12. In their report, along the equator in the Atlantic Ocean, temperature/salinity profiles of GLORYS12V1 are consistent with the observation (RMSE generally smaller than 0.4°C/0.3 psu in the water column). What should be pointed out here is that compared with the earlier version of GLORYS12 data, GLORYS12V1 has assimilated seasonal in-situ T-S profiles. Certainly, in the revised manuscript, those data qualities have been briefly given in the data section for clarification (line 68-69).

- **2. Several sentences are hard to follow and require revision and rephrasing. Some of these sentences are mentioned in the specific comments, but I encourage the authors to carefully go through the manuscript again and to clarify those statements.**
-
- **3. A number of important statements and sentences are missing references. For some of these statements the authors provide no references at all, while some require additional references or the correct references (see below for more detailed comments). This will help to better outline the new insights from this study by clarifying which statements are based on previous studies and which are based on the authors' new results.**

Thank you for the comments. We have proofread the whole manuscript to improve the language and correct the technical errors. All the detailed technical comments below have been addressed and the modifications of the manuscript are all marked as red font.

- **4. Figure 6b: I find this a nice figure. It nicely shows where off-equatorial Rossby waves are excited that will then reflect at the western boundary into equatorial Kelvin waves. However, I am wondering if these figures would look different if considering seasonally averaged anomalies instead of annually averaged climatologies (Fig. 6) and anomalies (Fig. 7)? This could potentially better highlight the dynamics of the 2019 Atlantic Nino event in Fig. 7.**

Thank you for the comment. We have calculated the seasonally averaged energy flux over Sep., Oct. and Nov. which is the onset season for the 2019 event shown as below (Fig.10 in the revised manuscript). Indeed, the figure has presented a different horizontal distribution of energy flux from the annual one, giving a better demonstration for the energy sources for the event. For

the second mode (Fig. 10b), we have found that the energy flux originated from the western boundary is not excited by local forcing in the western basin (see the mismatch between the wind anomaly and the flux origin). Additionally, westward energy flux by reflected Rossby waves is found in the eastern basin suggesting that a strong Kelvin wave is excited in summer for the second mode (in agreement with Fig. 9b). Correspondingly, for the third mode, also different from the annual flux in Fig. 7c where the easterward flux dominates the whole basin, Fig. 10c has revealed the locally forced Kelvin waveguide which originates from the central basin (around 15°W) and transfers the energy to the eastern basin in the event season. The detailed description has been added in the revised manuscript (line 208-217).

[Figure]

Seasonal mean zonal energy flux and wind stress anomaly averaged over Sep., Oct., and Nov. (Figure 10 in the revised manuscript). Color shadings are the zonal energy flux. Contours are the zonal wind stress anomaly with the interval of 0.002 N/m2.

- **5. Figure 9: If there is local forcing of the third and fourth mode between 20°W and 0° in the ATL3 region, it would be interesting to examine where exactly these modes are excited. Could the authors examine the spatial origin of these modes and discuss what is forcing them?**

Thank you for the comment. As we have mentioned in the minor comment 4, In the new Fig. 10, we have depicted the horizontal distribution of wave energy flux as well as the zonal wind anomaly in the event season (Sep., Oct. and Nov.) to illustrate the source region for waves in each mode. Additionally, in Figure 8 (below), we have also added the zonal wind anomaly as contours in the XT diagram for comparisons with the waveguide. By checking the wind anomaly, the discrepancy of wave energy source between the 2nd and 3rd/4th modes is notable. That is, high-mode waves (in the third and fourth mode shown as Figure 8 and 10) are strongly associated with the locally equatorial forcing, but the second-mode waves is more possible to be affected by off-equatorial waves (see the strong westward wave energy flux from

the off-equatorial region and the mismatch between the wind anomaly peak and the wave energy source in Figure 10b). The detailed description and analysis for those updated figures have also been given in the revised manuscript. The detail can be found in line (172-175) for Fig. 8 and line (208-217) for Fig. 10 of the revised manuscript.

[Figure]

Figure 8 in the revised manuscript. Contours are the zonal wind stress anomaly with the interval of 0.002 N/m2.

**Technical corrections and minor comments:**

Thank you for the careful review. All the technical corrections have been addressed in the revised manuscript. Some minor comments may still require a response, which we then have listed in the following.

- **Lines 14-16: There are several mechanisms that can trigger Atlantic Nino events (Luebbecke et al., 2018; Valles-Casanova et al., 2020) with the Bjerknes feedback being of one them.**

     Thank you for the comment. We admit that regarding the onset of the Atlantic Niños, there are several mechanisms exerting their influence, e. g. the Atlantic Meridional Mode and ENSO can modulate the event, and the variability of equatorial deep jet may serve as another energy source for the SST anomaly in the eastern Atlantic basin. However, none of them can individually trigger the Atlantic Niños without Bjerknes feedback. Including the studies by Luebbecke et al. (2018) and Valles-Casanova et al. (2020), those mechanisms were just introduced to provide the supplementary for the Bjerknes feedback to explain the diversity of the events.

- Line 15: "eastern" instead of "east".
- Line 20: I believe Prigent et al. (2020) is the correct reference here, instead of Crespo et al. (2022), for showing the reduction of interannual SST variability since 2000. The study of Crespo et al. (2022) focuses on projected changes of Atlantic Nino variability in CMIP6 models.
- Line 27: What exactly do the authors mean by "warning system"?
- Line 30: Remove "in" before "(Richter et al., 2022)".
- Lines 31-32: Please add a reference for this statement.
- Lines 37-38: What do the authors mean with vertical wave energy transfer that takes one month to reach the surface? Wind-forced KWs/RWs are excited at the surface and would transfer energy downward. Are the authors implying a wave forcing mechanism in the deep ocean?
- Line 39: "an" instead of "a".
- Line 44: "scheme" instead of "schemes".
- Line 45: "is" instead of "are".
- Line 80: Following the authors' notation, would it not be consequent to also denote the sea level anomaly with a prime? h' instead of h?
- Line 81: "sides" instead of "side".
- Line 94: "an offset term" instead of "a offset term".
- Line 97: Do the authors mean "through" instead of "though"?
- Lines 101-103: A reference like Cane & Moore (1981) or Brandt et al. (2016) is needed here.
- Lines 103-105: This sentence is hard to understand and needs rephrasing.
- Line 109: "includes" instead of "include".
- Line 109: What is meant here with instability waves?
-
- Lines 109-121: The results on climatological geopotential in Figure 3 are hardly described at all in this paragraph. A more detailed description of the results would be helpful.

Thank you for the comment. We have rewritten the whole paragraph (line 116-128) for more detailed description.

-
  - Lines 113-115, lines 118-121: Again, these sentences are hard to follow. Please rephrase.
  - Line 122: What is meant here with "features"? It would help to be more precise and to avoid such terms.
  - Line 123: "are" instead of "is".
  - Line 124: Better "deepening" than "drop".
  - Lines 124-127, lines 133-134: Meridional velocity seems very noisy and at very low levels (0.8 cm/s). I don't understand how the authors see/conclude sign-alternating behavior along the equator? The conclusion of mixed Rossby-Gravity waves in this discussion is rather speculative and not really based on the presented results. Either the authors should provide clearer evidence or consider removing this part.

Thank you for the comment. We can get the conclusion from the figure below, of which the color shading clearly demonstrates the sign-alternating behavior of the meridional velocity. We

understood that the contour in Fig. 5 is not as explicit as the color shading, however it did not violate the conclusion and we do not want to add extra figures for just presenting the meridional velocity, which is not really related to our main point.

[Figure]

XT-diagram of meridional velocity anomaly at the equator in 2019.

- **Line 127: Gravity waves instead of inertial waves on the equator?**
- **Line 129: Add a bracket after "see Figure 5".**
-
- **Line 141: Please be more precise. Where do the authors see strong eastward energy flux in Fig. 6a? It is north and south of the equator in the western basin and to a lesser degree on the equator in the eastern basin.**

Thank you for the comment. The "strong eastward energy flux" here is relative to the results from (Song and Aiki, 2020). We agree that this sentence may cause confusion.

Hence we have already revised it as "There is eastward energy flux that originates from the western boundary almost passing through the whole basin in the first mode (see Figure 6a). This eastward energy flux in the eastern basin and its connection with western boundary have not been seen in the research with linear ocean models" (line 149-151).

Song, Qingyang, and Hidenori Aiki. "The climatological horizontal pattern of energy flux in the tropical Atlantic as identified by a unified diagnosis for Rossby and Kelvin waves." *Journal of Geophysical Research: Oceans* 125.2 (2020): e2019JC015407.

- **Line 161: Remove one "the".**
- **Lines 161-162: "likely eliminates" instead of "is likely eliminate".**
- **Line 162: "occurrence" instead of "occur".**
- **Line 176: "locally" instead of "local".**
- **Lines 180-181: I don't understand why the authors say that the westward energy flux at S1 for the second mode peaks in September? Figure 9b shows maximum westward energy flux at S1 in January, February and June?**

Thank you for the comment. The local flux in Fig. 9 is determined by both the Kelvin and Rossby waves passing the transaction. Therefore, to explain the flux variation in S2, we should go back to Fig. 8 to check the energy transfer route. In Fig. 8b, it is found that Kelvin waves are holding from the boreal summer, meanwhile  in late June, a strong Rossby wave is just approaching S2 bringing the negative (westward) flux to pass S2 in Fig. 9b. In this sense, the reflected Kelvin waves by off-equatorial waves passing S1may indeed cause positive (eastward) flux in S2 from summer, however it is balanced by the negative (westward) flux of equatorial Rossby waves until Sep..  But we agree that the original sentence can cause confusion. Hence we have revised this sentence as "*Moreover, in the second mode, the eastward energy flux peaks in around Oct. on S2 just after strong westward energy flux passing the off-equatorial transection S1 from Jun. to Oct., which may suggest a wave energy transfer route that sequentially passes S1, S2 and S3 to influence the ATL3 region. It hence illustrates the influence of the wave energy from off-equatorial regions on the Atlantic Niño in 2019 to some extent.*" (line 197-200 of the revised manuscript).

- **Line 186: "recently" instead of "recent".**
- **Lines 188-190: Figure 4 shows that BCM4 is fairly low and close to zero during the 2019 Atlantic Nino event?**
- **Line 192: "propagation" instead of "travelling".**
- **Line 195: Figure 6a does not show such a pronounced westward energy flux as Figure 6b. How do the authors conclude that both modes are affecting westward Rossby waves?**

Thank you for the comment. We agree that  Figure 6a was not as evident as Figure 6b to show the westward energy flux. However the conclusion we got is not merely from Figure 6a, indeed we also checked Figure 9 where the westward energy flux passing S1 in the first mode is as strong as in the second mode (see Figure 9 a&b). On the other hand, since the first-mode equatorial wave has a longer Rossby deformation radius, its influenced latitude range is also extended. Hence, in Figure 6a, we should also focus on the westward flux in the latitude higher than 10ºN rather than the flux only between 0-10ºN as the second mode in Figure 6b. We have already indicated all the related figures in this sentence and additionally given a short explanation for Figure 6a in the related paragraph as "*Thus we found a broader latitude coverage of westward energy flux in lower modes which may suggest the possible off-equatorial RWs (e.g. the westward energy flux within 15º in the north for the first mode and within 10º for the second mode)*" (line 147-149 ) for clarification.

- **Line 199: "demonstrate" instead of "demonstrated".**
- **Lines 199-202: Another sentence that is very hard to follow. Please rephrase.**
- **Line 202: "research" instead of "researches".**
- **Line 213: This statement requires a reference. Which study concludes that equatorial waves provide great potential to predict Atlantic Ninos?**
- **Figure 2 caption: Remove one "by" in the second line.**
- **Figure 4: Please add a label to the y axis.**

Thank you for the comment. Values for all the variables shown in Figure 4 are indeed normalized (see the caption), therefore those variables are dimensionless.  In this sense, "variation" might be a proper y-label and has been added in the updated figure.

References

- Brandt, P., Claus, M., Greatbatch, R. J., Kopte, R., Toole, J. M., Johns, W. E., & Böning, C. W. (2016), Annual and Semiannual Cycle of Equatorial Atlantic Circulation Associated with Basin-Mode Resonance. *Journal of Physical Oceanography*, 46, 3011-3029, https://doi.org/10.1175/JPO-D-15-0248.1
-
- Brandt, P., Hahn, J., Schmidtko, S., Tuchen, F. P., Kopte, R., Kiko, R., Bourlès, B., Czeschel, R., & Dengler, M. (2021), Atlantic Equatorial Undercurrent intensification counteracts warming-induced deoxygenation. *Nature*
- *Geoscience*, 14, 278-282, https://doi.org/10.1038/s41561-021-00716-1
-
- Cane, M. A., and D. W. Moore (1981), A Note on Low-Frequency Equatorial Basin Modes. *Journal of Physical Oceanography*, 11, 1578-1584, https://doi.org/10.1175/1520-0485(1981)011,1578:ANOLFE.2.0.CO;2
-
- Lübbecke, J. F., Rodríguez-Fonseca, B., Richter, I., Martín-Rey, M., Losada, T., Polo, I., & Keenlyside, N. S. (2018), Equatorial Atlantic variability – modes, mechanisms, and global teleconnections. *WIREs Climate Change*, 9:e527. https://doi.org/10.1002/wcc.527
-
- Perez, R. C., Foltz, G. R., Lumpkin, R., & Schmid, C. (2019), Direct Measurements of Upper Ocean Horizontal Velocity and Vertical Shear in the Tropical North Atlantic at 4°N, 23°W. *Journal of Geophysical Research: Oceans*, 124, 4133-4151. https://doi.org/10.1029/2019JC015064
-
- Prigent, A., Lübbecke, J., Bayr, T., Latif, M. & Wengel, C. (2020), Weakened SST variability in the tropical Atlantic Ocean since 2000. *Climate Dynamics*, 54, 2731–2744. https://doi.org/10.1007/s00382-020-05138-0
- Tuchen, F. P., Perez, R. C., Foltz, G. R., Brandt, P., & Lumpkin, R. (2022a), Multidecadal Intensification of Atlantic Tropical Instability Waves. *Geophysical Research Letters*, 49, e2022GL101073. https://doi.org/10.1029/2022GL101073
- Tuchen, F. P., Brandt, P., Hahn, J., Hummels, R., Krahmann, G., Bourlès, B., & Coauthors (2022b), Two Decades of Full-Depth Current Velocity Observations From a Moored Observatory in the Central Equatorial Atlantic at 0°N, 23°W. *Frontiers in Marine Science*, 9:910979. https://doi.org/10.3389/fmars.2022.910979

Vallès-Casanova, I., Lee, S.-K., Foltz, G. R., & Pelegrí, J. L. (2020), On the spatiotemporal diversity of Atlantic Niño and associated rainfall variability over West Africa and South America. *Geophysical Research Letters*, 47, e2020GL087108. https://doi.org/10.1029/2020GL087108

---

## Author Response (AR2)

**Response to the reviewer**

Thank you very much for providing the valuable feedback again on our manuscript. All the technical corrections below have been addressed. In the following are the point-to-point responses to the remaining comments.

**Major comments:**

**1) Thanks for mentioning and referring to the quality check that has been carried out by the CMEMS team. I agree that the mean currents seem to be represented when compared to vertical mean profiles from moored observations. However, my concern was not so much about the representation of the mean currents, but mainly about the variability of the currents which is a considerably bigger challenge for reanalysis products (Tuchen et al., 2022). At least some discussion on the uncertainty of velocity variability on Kelvin wave time scales is needed here.**

Thank you very much for this comment. By carefully reading the paper by Tuchen et al., 2022, we agree that a short discussion about the reliability of reanalysis dataset is necessary, which we think should be put in the Summary section (line 261-265) as one limitation of our diagnosis scheme.

**2) Thanks for explaining the difference between AGC flux from a climatological and from an anomalous perspective. I think what was confusing to me were the lines of theoretical RW and KW propagation in Fig. 8 (which still do not fit very good, neither are easily to detect as the authors write). Why do all KW lines start at the beginning of the year? Should this not be fitted to the actual flux pattern?**

Thank you for the comment. In 2019, the KW train does start at the year beginning, which indeed has been presented with linear ocean models by Song et. al (2023). In their models, the KW is likely to be reflection-induced by RWs rather than local winds. As we have mentioned in the last response letter, the real problem in Fig. 8 is the mixture of KW signals at multiple frequencies. We think a low-pass filter should be helpful to improve the figure presentation especially for RWs, however in this study, we do not want to manipulate the extracted signals too much.

Song, Q., Tang, Y., & Aiki, H. (2023). Dual Wave Energy Sources for the Atlantic Niño Events Identified by Wave Energy Flux in Case Studies. *Journal of Geophysical Research: Oceans*, *128*(7), e2023JC01997

**Remaining technical corrections and minor comments:**

**- Lines 13-14: Atlantic Nino is associated with positive SST anomalies, while Atlantic Nina is associated with negative SST anomalies. Here, it should be made clear that only positive SST anomalies are referred to Atlantic Nino events (as part of the Atlantic zonal mode with its negative and positive phases). (HAS NOT BEEN ADDRESSED IN THE REVISED VERSION)**

Thank you for the comment. We have clarify this in Lines 13- 14 as " The equatorial Atlantic Ocean is known for exhibiting pronounced anomalies of sea surface temperature (SST) on interannual time scales, of which the events with positive anomalies are often referred to as Atlantic Ninos"

**- Line 113: What is the motivation for mentioning tropical instability waves here without any further explanation? Is the near-equatorial wind-forced wave signal considerably disturbed by TIWs?**

Thank you for the comment. The reason we mentioned TIW is to emphasize that the extracted variability will not only include the wave signal due to wind forcing. However we agree that putting TIW here may confuse readers. In the revised manuscript, we have removed this sentence.

**- Line 135: Gravity waves instead of inertial waves on the equator? (HAS NOT BEEN ADDRESSED IN THE REVISED VERSION)**

Thank you for the comment. We have changed "inertial waves" to "gravity waves".

**- Lines 222: "All" instead of "all".**

Thank you for the comment. We have corrected the typo.

**- Line 260: This statement requires a reference. Which study concludes that equatorial waves provide great potential to predict Atlantic Ninos? (HAS NOT BEEN ADDRESSED IN THE REVISED VERSION)**

Thank you for the comment. We have added three references, of which (Imbol Koungue et. al 2017) proposed to implement a linear ocean model to simulate equatorial waves for warning Atlantic Nino and Benguela Nino, and (Song et. al 2023b) and (Richter et. al, 2022) presented how equatorial waves are working on the onset of the 2019 event.

Imbol Koungue, R. A., Illig, S., & Rouault, M. (2017). Role of interannual K elvin wave propagations in the equatorial A tlantic on the A ngola B enguela C urrent system. *Journal of Geophysical Research: Oceans*, *122*(6), 4685-4703.

Song, Q., Tang, Y., & Aiki, H. (2023). Dual Wave Energy Sources for the Atlantic Niño Events Identified by Wave Energy Flux in Case Studies. *Journal of Geophysical Research: Oceans*, *128*(7), e2023JC01997

Richter, I., Tokinaga, H., & Okumura, Y. M. (2022). The extraordinary equatorial Atlantic warming in late 2019. *Geophysical Research Letters*, *49*(4), e2021GL095918.